# Is there a relationship between isometric hamstrings-to-quadriceps torque ratio and athletes' plyometric performance?

**Farideh Babakhani**⊙*, **Mohamadreza Hatefi, Ramin Balochi**

Department of Sport Injuries and Corrective Exercises, Faculty of Physical education, Allameh Tabataba'i University, Tehran, Iran

* Farideh_babakhani@yahoo.com

## Abstract

### Background

The application of the modified reactive strength index ($RSI_{mod}$) and isometric hamstrings to quadriceps (H:Q) torque ratio to monitoring the athletes' performance is well established, but their relationship to each other still remains unknown. Therefore, the purpose of this investigation was to clarify whether there is a relationship between $RSI_{mod}$ and the isometric H:Q torque ratio.

### Methods

Twenty-one male recreational athletes (age, 24.89 ± 4.46 years; weight, 74.11 ± 8.66 kg; height, 179.78 ± 6.76 cm) volunteered to participate in this research. Their isometric H:Q torque ratio via hand-held dynamometer and jumping performance variables during the stop jump (STJ), double leg-drop vertical jump (DL-DVJ), and single leg-drop vertical jump (SL-DVJ) tasks were measured. Also, the $RSI_{mod}$ was calculated by dividing the vertical jump height by the time to take-off. Pearson correlation coefficients were used to determine the relationship between the $RSI_{mod}$ and jumping performance variables.

### Results

The result showed a significant strong positive relationship between the H:Q torque ratio and STJ performance based on the $RSI_{mod}$ ($p$ = .027, r = .724). Although, there was a moderate positive relationship between the H:Q torque ratio and jumping height of the STJ task, but it wasn't statistically significant ($p$ = .096, r = .588). Also, no significant relationship was found between the H:Q torque ratio and all jumping performance variables of the DL-DVJ and SL-DVJ tasks ($p \geq$ .05).

### Conclusion

The current study exhibited that the isometric H:Q torque ratio correlates to STJ performance based on $RSI_{mod}$ but not to SL-DVJ and DL-DVJ. Notably, the difference in results between jumping tasks can be attributed to the complexity of the movement, which means that the $RSI_{mod}$ is probably related to other factors. Therefore, the isometric H:Q torque ratio

**Data Availability Statement:** All relevant data are within the paper and its Supporting Information files.

**Funding:** The authors received no specific funding for this work.

**Competing interests:** The authors declare that they have no conflicting interests.

used to monitor the athletes' performance couldn't independently represent the jumping performance that is determined by $RSI_{mod}$.

## Introduction

Muscle strength is one of the key components of athletes' performance, and in many sports due to the necessity of performing maximal repeated jumps during the activities, it has been shown that there is a direct relationship between the strength of the knee extensor muscles and the jumping performance and subsequently sport successes [1–3]. In this regard, various exercises have been designed to increase knee extensor strength, and the effectiveness of these exercises is often measured by using different methods of muscle strength measurement, directly by an isokinetic dynamometer and hand-held manual dynamometer (HHD), as an inexpensive, portable, and more clinically applicable devices, or indirectly by using functional movement tests such as jump-landing tasks. On the other hand, it has been well proven that muscle strength should be considered as an important factor in reducing the risk of non-contact injuries among athletes, and improving it has been also recommended as a component of an injury prevention program [4].

In movement tasks with high-velocity conditions including jump-landing, the quadriceps muscle plays an important role to produce explosive force and also controlling the knee flexion acceleration with the collaboration of the hamstring muscle as a hip extensor, and also reducing the shearing force on the anterior cruciate ligament (ACL) and controlling hip flexion and knee extension accelerations [5]. Deficient in the hamstrings and quadriceps muscles strength, as well as the change in their strength ratio to each other (as an indicator of the strength balance between the antagonistic muscle groups (extensor and flexor) around the axes of knee rotation) has been identified as a modifiable intrinsic risk factor for ACL injuries and hamstring strains; Which should be considered in injury prevention programs, and subsequently, if necessary, the hamstring (H)-to-quadriceps (Q) torque ratio (H:Q) should be improved [6–8]. For this reason, in addition to muscle strength, or more specifically the ability to produce torque (force times the moment arm) by muscles, the H:Q torque ratio is considered an integral part of evaluating the athletes' performance or monitoring the rehabilitation process of people with various injuries [6].

In contrast, the modified reactive strength index ($RSI_{mod}$) is introduced as a valid index to evaluate the athletes' plyometric performance as well as neuromuscular readiness, which is calculated by dividing the vertical jump height by the time to take-off [9, 10]. $RSI_{mod}$ is defined as the ability to quickly and efficiently change eccentric contraction to concentric contraction, which is used to measure the lower extremity explosive power for evaluating the athletes' plyometric performance [11]. Basically, using the $RSI_{mod}$ is considered vital for high-performance athletes, because it can be used as a motivational tool by providing the athletes with immediate feedback about their own achieved $RSI_{mod}$ score, and if necessary improving their movement performance with a skill and injury prevention approach during training programs [12]. Actually, the $RSI_{mod}$ is used 1) as a practical method to quantify the stretching-shortening cycle during jump-landing movement tasks [13], 2) to monitor the quality of training of sports teams [14], and 3) or as a diagnostic test to evaluate the functional ability of subjects with ACL deficient [15]. In this regard, it has been reported that there is a direct relationship between extensor muscle strength [16], triple hop performance test [3], change-of-direction speed [17], and agility [18] with the $RSI_{mod}$. However, no study was found that evaluate the relationship between isometric H:Q torque ratio and athletes' plyometric performance by using $RSI_{mod}$.

In summary, both $RSI_{mod}$ and the H:Q torque ratio variables as performance criteria are considered for monitoring athletes with the approaches of movement performance, injury prevention, and returning to sports after an injury, or evaluating the effectiveness of training interventions in athletes with different conditions; But there is no evidence if these two variables are related to each other. Therefore, the current research question is whether there is an relationship between the isometric H:Q torque ratio and the jumping performance variables of athletes or not? Notably! we hypothesized that isometric H:Q torque ratio would be positively correlated with jumping performance.

## Methods

### Participants

According to G. Power software version 3.1.0 (Franz Faul, University of Kiel, Germany), based on a Pearson correlation statistical test and assuming a power of 0.80, an effect size of 0.8, and a two-tailed alpha level of 0.05, twenty-one male recreational athletes (10 football, 6 basketball, 4 volleyball, and 1 handball players) were voluntarily participate to this investigation (age, 24.89 ± 4.46 years; weight, 74.11 ± 8.66 kg; height, 179.78 ± 6.76 cm); Which were selected according to the study criteria. Participants were recruited for the current study through the board of the faculty of physical education and sports sciences since May 1, 2023, for a month. In this study, a recreational athlete was defined as a subject who participates in sports activities at least three times a week for at least 30 minutes. Inclusion criteria were to: be active physically, be aged 18 to 30 years, have a body mass index (BMI) between 18 and 24, and have a normal ankle dorsiflexion range of motion at least 20˚ based on the ankle lunge test [19]. Participants were excluded if they: had any musculoskeletal injury in the previous two months or lower-extremity injury in the previous six months, had a lower limb surgery or fractures within the past year, and had any neurological and pathological conditions.

Prior to the test, ethical approval was obtained by the ethical committee of Allameh Tabataba'i University (IR.ATU.REC.1402.008), and all participants provided written informed consent.

### Procedures

In the present study, participants were referred to the athletic training laboratory on one occasion and completed a single half-hour testing session. They were asked to wear comfortable sports clothing and their own sport shoes. Overall, first, the strength of the quadriceps and hamstring muscles of the participants was measured by an HHD, and after that, they were asked to perform the stop jump (STJ), double leg-drop vertical jump (DL-DVJ), and single leg-drop vertical jump (SL-DVJ) tasks on a force plate with 1 min rest between two consecutive repetitions and 2 min rest between tasks. Also, a 5 min rest was considered between the muscle strength test and performance tests. Before performing the strength test, the participant performed a 5 min warm-up routine of stretching and aerobic exercises with moderate intensity. Also, they were asked to perform the jumping tasks before data capturing for familiarization.

### The hamstring-to-quadriceps (H:Q) torque ratio measurement

The quadriceps and hamstring muscle strength of the dominant leg was measured by HHD as a valid and reliable device [20, 21] (Nicholas Manual Muscle Tester, Lafayette Instrument Company, Lafayette, Indiana, USA) and then the recorded maximum value was multiplied by the distance between the center of the dynamometer and the rotation axis of the knee; Which was measured by tape measure. To measure the isometric hamstring strength, the participants

were asked to lay in a prone position with knees and hips in 90˚ flexion and hands freely behind the body. In the next step, the HHD was placed 2 cm above the lateral malleus in the posterior part of the tibia, and the participants were asked to produce maximum force for 5 s in the knee flexion direction. To measure the isometric quadriceps strength, the participants were asked to sit on the table with knees and hips in 90˚ flexion and hands crossed over the chest. In the next step, the HHD was placed 2 cm above the lateral malleus in the anterior part of the tibia, and the participants were asked to produce maximum force for 5 s in the knee extension direction.

It should be noted that each test was performed for two trials with a 30-s resting period interval between them and a 2-min between muscles, and the highest value was considered as the maximum isometric strength; Notably, the participants were verbally encouraged to perform the maximum effort during the test, and if the difference between two measurements was more than 10%, the test was repeated. Also, if pain and discomfort are reported during the test, the test was stopped and repeated.

Finally, the quadriceps and hamstrings muscles torque (Nm/kg) was calculated by multiplying the force, acceleration of gravity, and lever arm: [force (kg) × 9.81 × lever arm (m)] and normalized by the body mass (kg) of each subject. Also, isometric H:Q torque ratio for each participant was calculated by dividing hamstring torque by the quadriceps torque multiplied by 100.

## Jump tasks procedures

To perform the STJ task, the participants were asked to run towards the force plates, quickly decelerate and jump with both feet on the force plate, then immediately perform a two-footed maximal vertical jump (Fig 1A). To perform the DL-DVJ task, first, the participants were asked to stand with feet shoulder-width apart on the 30-cm height step which was placed at a distance equal to half of the subject's height in relation to the center of force plate; Then, they were asked to land with both feet on a force plate, and immediately perform a maximal vertical jump as fast as they can (Fig 1B). To perform the SL-DVJ task, the participants were asked to do the same task as the DL-DVJ with the difference that landing and jumping must be done with the dominant leg (Fig 1C); The dominant leg was defined as the leg that was chosen in the sudden landing.

## The modified reactive strength index ($RSI_{mod}$) measurement

$RSI_{mod}$ is used as a method to measure the lower extremity explosive strength to assess the athletes' jumping performance. $RSI_{mod}$ was calculated by dividing the vertical jump height by the time to take-off which indicates the athlete's jump ability relative to the duration of the applied force or time to take-off [22]; Jump height was calculated by the vertical force-time graph of the force plate device sampling at 1200 Hz based on this formula: [9/81× (flight time)^2) /8]; Time to take-off was defined as the duration length of force-time between the landing and take-off (vertical force threshold was defined as 10% of the subject's body weight by using MATLAB software) [22–24]. Notably, the $RSI_{mod}$ has been shown to be a reliable and valid measure [22], a practical way to assess the lower limb explosive performance [13].

## Statistical analysis

Regarding the normality of data distribution based on the Shapiro-Wilk test, the Pearson correlation coefficient statistical test was used to determine the relationship between the H:Q torque ratio and jumping performance. All data were calculated by use of the SPSS software Version 22.0 (Microsoft Corp., Redmond, WA), and the significance level was set at .05.

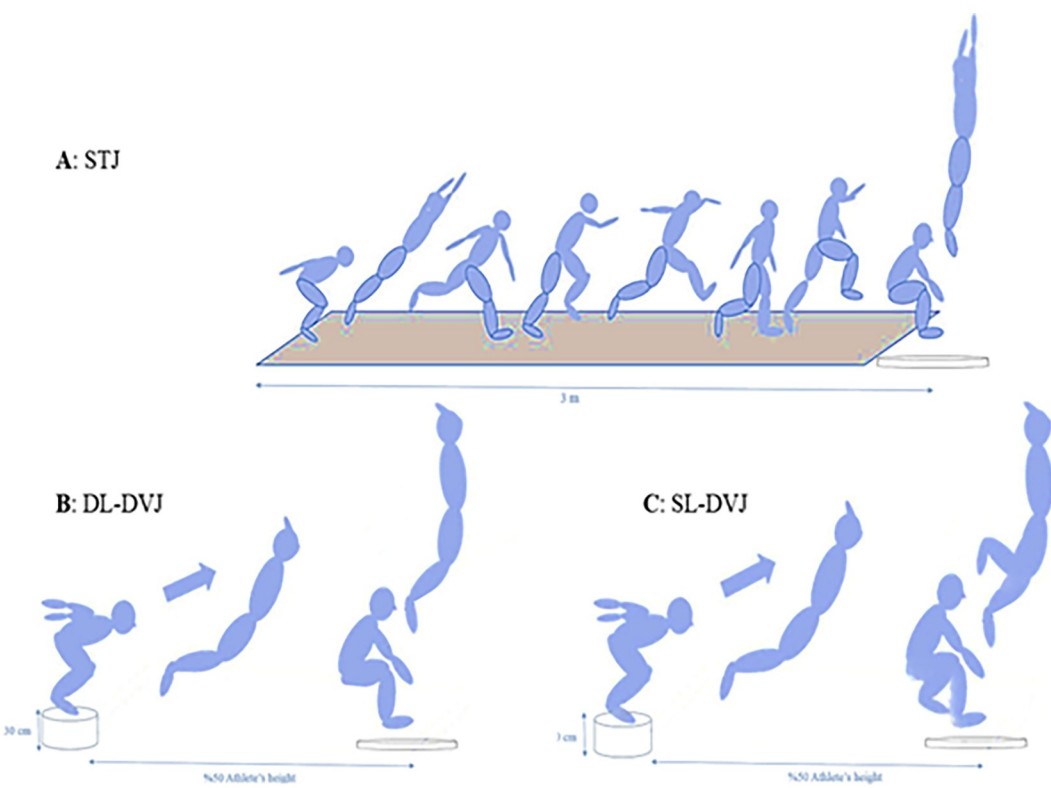

**Fig 1. The illustration of the method of the jumping tests.** A: Stop jump (STJ), B: Double leg-drop vertical jump (DL-DVJ), and C: Single leg-drop vertical jump (SL-DVJ).

## Results

The mean and SD values for jumping performance tests' variables and the hamstring-to-quadriceps strength ratio of participants can be found in Table 1. The result of the Pearson correlation coefficient test showed a significant strong positive relationship between the H:Q torque ratio and STJ performance based on the $RSI_{mod}$ ($p = .027$; $r = .724$ [95% CI, .446 to .946]). Although, there was a moderate positive relationship between the H:Q torque ratio and jumping height of the STJ task, but it wasn't statistically significant ($p = .096$; $r = .588$ [95% CI, .188 to .869]). Also, no significant relationship was found between the H:Q torque ratio and all jumping performance variables of the DL-DVJ and SL-DVJ tasks (Table 2).

**Table 1. Jumping performance tests' variables values and the hamstring-to-quadriceps (H:Q) torque ratio score of participants[^].**

| variables | | JH (m) | CT (s) | $RSI_{mod}$ (m/s) |
|---|---|---|---|---|
| **Jumping tasks** | **DL-DVJ** | .28 ± .06 | .43 ± .09 | .68 ± .20 |
| | **SL-DVJ** | .16 ± .04 | .42 ± .10 | .43 ± .15 |
| | **STJ** | .31 ± .06 | .35 ± .06 | .94 ± .18 |
| **H:Q torque ratio (Nm/kg)** | | 76.05 ± 11.96 | | |

[^]Data are presented as mean ± SD. *Abbreviations*: JH, jump height; CT, ground contact time; $RSI_{mod}$, modified reactive-strength index; DL-DVJ, Double leg- drop vertical jump; SL-DVJ, Single leg- drop vertical jump; STJ, Stop jump.

**Table 2. Correlation (r) between isometric hamstring-to-quadriceps (H:Q) torque ratio and jumping performance.**

| Variables | | | JH | CT | $RSI_{mod}$ |
|---|---|---|---|---|---|
| H:Q torque ratio | DL-DVJ | r (95% CI) | .435 (-.184 to .930) | -.185 (-.751 to .427) | .459 (.017 to .803) |
| | | p-value | .242 | .634 | .214 |
| | SL-DVJ | r (95% CI) | .187 (-.607 to .814) | -.206 (-.790 to .614) | .269 (-.262 to .716) |
| | | p-value | .630 | .595 | .484 |
| | STJ | r (95% CI) | .588 (.188 to .869) | .111 (-.536 to .670) | .724* (.446 to .946) |
| | | p-value | .096 | .776 | .027 |

* Correlations significant at $p \leq .05$. *Abbreviations*: JH, jump height; CT, ground contact time; $RSI_{mod}$, modified reactive-strength index; DL-DVJ, Double leg- drop vertical jump; SL-DVJ, Single leg- drop vertical jump; STJ, Stop jump; 95% CI, 95% Confidence interval.

## Discussion

Several factors including extensor muscle strength, triple hop performance test, change-of-direction speed, and agility are related to athletes' jumping performance based on $RSI_{mod}$ [3, 16, 18]; Notably, there is some study that evaluated the relationship between dynamic/mixed type of the H:Q torque ratio and $RSI_{mod}$ in various tasks with different conditions [25–27]. Although, the gold standard method to quantify muscle strength utilizes an isokinetic dynamometer, However, this option lacks clinical applicability due to cost and size. HHD is a portable and more clinically applicable device well establishes as a valid and reliable testing alternative. To date, the relation of the isometric H:Q torque ratio, especially in various jumping tasks has not been well investigated. The purpose of this study was to determine the relationship between the isometric H:Q torque ratio and jumping performance variables. Our results indicated that the isometric H:Q torque ratio significantly had a strong positive correlation with $RSI_{mod}$ of the STJ task. But, no significant relationship was found between the isometric H:Q torque ratio and all jumping performance variables of the DL-DVJ and SL-DVJ tasks. Although, the relationship between the H:Q torque ratio and the $RSI_{mod}$ of the DL-DVJ task wasn't statistically significant, but, there was a moderate positive relationship between them; Notably, the correlation between the H:Q torque ratio and the $RSI_{mod}$ of the SL-DVJ task was poor; These results suggest that the $RSI_{mod}$ can be attributed to a much more complex process than to H:Q torque ratio, especially in tasks with high complexity, including SL-DVJ task.

In this regard, recent similar studies have shown that the correlation of the H:Q torque ratio with jumping performance variables is different in various tasks with different conditions, which means that it could be related to movement task demand [26, 27]; Diker et al. demonstrated that, although, the H:Q torque ratio was correlated with 30 m sprint time but it wasn't with countermovement jump or squat jump height tasks [26]; Struzik and Pietraszewski showed a different correlation of the H:Q torque ratio with jumping performance variables by changing the velocity changes-based isokinetic H:Q torque ratio or box height of the drop jump task [27]. It is worth noting that the difference between these studies and our study was that they defined jump height as a jumping performance and also measured the dynamic/mixed type of the H:Q torque ratio. According to the finding of the current study and considering the results of previous similar studies in regard to the relationship between the H:Q torque ratio and jumping performance in different task conditions, the authors hypothesize that as much as the complexity of the movement is increase, the correlation between H:Q torque ratio and $RSI_{mod}$ decreases; In other words, the jumping performance rate is probably more

related to other factors. In this context, various studies well have proven that the physical demands are not the same in different movement tasks or different conditions which can be attributed to the results of the current study [28–31]. Therefore, the isometric H:Q torque ratio used to monitor the athletes' performance couldn't independently represent the jumping performance that is determined by $RSI_{mod}$. Authors recommend that monitoring strength balances along with other factors may provide a better understanding of the association between H:Q torque ratio and jumping performance variables.

In summary, according to the strong correlation of the isometric H:Q torque ratio with the $RSI_{mod}$ of the STJ task, calculated $RSI_{mod}$ during the STJ task, but not during DL-DVJ and SL-DVK tasks, can be used as an additional tool for the recognition of the isometric H:Q torque ratio. However, it should be noted that even in this case, we investigated the isometric H:Q ratio, and this correlation value might have been different if the H:Q ratio was calculated based on the dynamic or mixed type of ratio. In this regard, some research has shown a significant association between a different type of H:Q torque ratio and injuries [6]. Also, it is reported that the dynamic strength tests may be more practical methods of assessing the relationships between relative strength levels and dynamic performance especially in athletes [32]. Although, in this regard, it has been shown that the dynamic H:Q torque ratio is also related to jumping performance based jumping height and should be considered in strength-based training programs which was similar to our finding [33]. However, it seems that a comprehensive study is needed to investigate the dynamics H:Q torque ratio on jumping performance during different jumping tasks.

We acknowledge that the current study had limitations that should be considered; First, we used the isometric H:Q torque ratio, and the other type of ratio may have a different correlation with $RSI_{mod}$; Second, the participants in this study were healthy male recreational athletes, so these results may not be generalizable to everyone; Gender differences may play an important role in jumping movements pattern and, of course, the results may have been affected if athletes of different sports were evaluated.

## Conclusion

In summary, we investigated the relationship between isometric H:Q torque ratio and athletes' jumping performance; The findings of the current study indicated that the isometric H:Q torque ratio correlates to STJ performance based on $RSI_{mod}$ but not to SL-DVJ and DL-DVJ. Notably, the difference in results between jumping tasks can be attributed to the complexity of the movement, which means that the $RSI_{mod}$ is probably related to other factors. Therefore, the isometric H:Q torque ratio used to monitor the athletes' performance couldn't independently represent the jumping performance that is determined by $RSI_{mod}$.

## Supporting information

**S1 Data.**
(XLSX)

## Acknowledgments

The authors would like to thank all participants and athletic training laboratory staff for the collaborations to make this study.

## Author Contributions

**Conceptualization:** Farideh Babakhani, Ramin Balochi.

**Data curation:** Farideh Babakhani, Mohamadreza Hatefi, Ramin Balochi.

**Investigation:** Farideh Babakhani, Mohamadreza Hatefi.

**Software:** Mohamadreza Hatefi.

**Supervision:** Farideh Babakhani.

**Visualization:** Ramin Balochi.

**Writing – original draft:** Farideh Babakhani, Mohamadreza Hatefi.

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
