## [Decision Letter · Decision Letter 0]

9 Aug 2023

PONE-D-23-21774Relationship Between Isometric Hamstrings to Quadriceps Torque Ratio and Athletes' Plyometric PerformancePLOS ONE

Dear Dr. Babakhani,

Thank you for submitting your manuscript to PLOS ONE. After careful consideration, we feel that it has merit but does not fully meet PLOS ONE’s publication criteria as it currently stands. Therefore, we invite you to submit a revised version of the manuscript that addresses the points raised during the review process.

We look forward to receiving your revised manuscript.

Kind regards,

Esedullah Akaras

Academic Editor

PLOS ONE

Journal Requirements:

Additional Editor Comments:

Dear authors, a major revision has been requested for your article.

Reviewers' comments:

Reviewer's Responses to Questions

**Comments to the Author**

1. Is the manuscript technically sound, and do the data support the conclusions?

Reviewer #1: Partly

Reviewer #2: Yes

2. Has the statistical analysis been performed appropriately and rigorously? 

Reviewer #1: Yes

Reviewer #2: Yes

3. Have the authors made all data underlying the findings in their manuscript fully available?

Reviewer #1: Yes

Reviewer #2: Yes

4. Is the manuscript presented in an intelligible fashion and written in standard English?

Reviewer #1: No

Reviewer #2: Yes

5. Review Comments to the Author

Reviewer #1: Dear Authors,

First of all, thank you for valuable effort on this manuscript.

The following points are presented after my review.

- The title of the study should be revised. The entire text has been academic language-reviewed.

- Is "double" meant to be denoted by "dabble" in the term "dibble leg-drop vertical jump"? The use of the word "dabble" does not seem appropriate.

- It is recommended to briefly indicate which parameters are used in the RSImod calculation in the method section of the summary.

- The H:Q ratio mentioned in the Introduction section is evaluated as isometric and more recently isokinetic tests (because it provides the opportunity to perform functional analysis). Within the scope of this study, the reasons why the isometric H:Q ratio will be examined instead of the dynamic H:Q ratio should be given in more detail.

- Although it is mentioned for what purpose the RSDmod index is used or what it is an indicator of, it has not been mentioned exactly how it is calculated. For a first-time reader, although methods are also mentioned, it would be better to include a brief explanation over a task in this section, at least by mentioning which physical parameters it is calculated using for this index.

- "If yes! to what extent does increasing or decreasing H:Q torque ratio affect the athletes' jumping performance?" Instead of this question, a interpretation can be made based on the according results, or not specified at all.

- Despite the power analysis, sample size is limited. Considering the inclusion of subjects of different genders, the number of subjects is low compared to other similar studies (usually single-gender subjects were evaluated in other similar studies) and in terms of making a clear conclusion from the results of the study.

- No information on gender distribution was provided.

- It is recommended to mention which sports the subjects are doing.

- Ethics committee approval protocol number should be attached.

- If the tests were done in a single session, it should be stated whether sufficient rest breaks are given between muscle test and performance tests and how long it is given.

- As a procedure explanation, which test has been done first, the procedure of this test should be given first, followed by the other. The relevant sections should be rearranged by paying attention to the order of the test procedure explanations. A subtitle can be created about jump tests, as others.

- The figures in figure 1 are illustrative but it has not clearly seen as one leg during landing on the SL-DVJ task. Is it the first landing on one leg or the second landing? or both of them? It is recommended to review and revise the figure from this perspective.

- Under the heading "The hamstring-to-quadriceps (H:Q) torque ratio measurement", the calculation part should be placed after the test procedure, not at the beginning of the paragraph.

- The quadriceps femoris muscle, generally referred to as "quadriceps" throughout the manuscript, should be corrected.

- The relationship between isokinetic H/Q ratio and jumping performance parameters was investigated without using RSDmod. Some recent studies are listed below. The discussion can be rearranged by reviewing these studies.

o Atik, B., Ayberk, B., Özgül, B., & Polat, M. G. (2023). The Association Between Isokinetic Strength and Strength Asymmetry and Jump Performance in Female Volleyball Players. Sport Sciences for Health, 1-8.

o Schons P, Da Rosa RG, Fischer G et al (2019) The relationship between strength asymmetries and jumping performance in professional volleyball players. Sports Biomech. 18(5):515–526.

- One of the evaluations applied in the study is a static measurement while the other is a dynamic task. Therefore, instead of a static measurement, comments can be added about the reflection of dynamic muscle strength (isokinetic) evaluations on RSDmod can be examined. Although this situation is partially mentioned at the end of the discussion, comparative comments can be made in the first parts of the discussion by giving place to case studies from the literature.

- It should be noted that RSDmod was not used in the studies mentioned in sources 23 and 24.

- Since the RSDmod index was not used in the mentioned studies, "According to the finding of the current study and considering the results of previous studies, the authors hypothesize that as much as the complexity of the movement is increase, the correlation between H:Q torque ratio and RSImod decreases " can be commented, it is recommended to review again.

- The results of the study may also have been affected by the evaluation of athletes doing different sports.

- The discussion generally seems rather inadequate. It should be rearranged by including current similar studies.

Best Regards

Reviewer #2: The subject raised by the authors is extremely interesting. The effect of the H/Q ratio on the risk of injury is quite extensively described. However, there are few studies describing the relationship between the H/Q ratio and athletes performance. Therefore, the presented manuscript is part of filling the current knowledge gap.

The first chapter adequately introduces the reader to the problem and shows the gap in knowledge to be filled.

The methods have been described quite extensively. However, authors should use a different abbreviation than SJ for "stop jump". The abbreviation SJ is adopted and used for squat jump. Therefore, one should not suggest to the reader that the article concerns a different, biomechanically standardized jump.

The results were described and presented in sufficient detail. However, specifying the RSI unit as m/s should be considered overzealous and unnecessary. We don't want to talk about movement velocity here.

The discussion includes the interpretation of the results and references to the work of other authors.

In the Discussion - I couldn't find an author named Dekker [24]... I think it was Diker, right?

The interpretation of the obtained results and the final conclusions seem to be correct.

6. PLOS authors have the option to publish the peer review history of their article (what does this mean?). If published, this will include your full peer review and any attached files.

Reviewer #1: No

Reviewer #2: No

---

## [Decision Letter · Decision Letter 1]

19 Oct 2023

PONE-D-23-21774R1Is there a Relationship Between Isometric Hamstrings-to-Quadriceps Torque Ratio and Athletes' Plyometric Performance?PLOS ONE

Dear Dr. Babakhani,

Thank you for submitting your manuscript to PLOS ONE. After careful consideration, we feel that it has merit but does not fully meet PLOS ONE’s publication criteria as it currently stands. Therefore, we invite you to submit a revised version of the manuscript that addresses the points raised during the review process.

We look forward to receiving your revised manuscript.

Kind regards,

Esedullah Akaras

Academic Editor

PLOS ONE

Journal Requirements:

Reviewers' comments:

Reviewer's Responses to Questions

**Comments to the Author**

1. If the authors have adequately addressed your comments raised in a previous round of review and you feel that this manuscript is now acceptable for publication, you may indicate that here to bypass the “Comments to the Author” section, enter your conflict of interest statement in the “Confidential to Editor” section, and submit your "Accept" recommendation.

Reviewer #1: (No Response)

Reviewer #2: All comments have been addressed

2. Is the manuscript technically sound, and do the data support the conclusions?

Reviewer #1: Yes

Reviewer #2: Yes

3. Has the statistical analysis been performed appropriately and rigorously? 

Reviewer #1: Yes

Reviewer #2: Yes

4. Have the authors made all data underlying the findings in their manuscript fully available?

Reviewer #1: Yes

Reviewer #2: Yes

5. Is the manuscript presented in an intelligible fashion and written in standard English?

Reviewer #1: Yes

Reviewer #2: Yes

6. Review Comments to the Author

Reviewer #1: Dear Authors,

Firstly thank you for your effort on the revisions of the manuscript. With the revisions made, it is now in a better condition.I would only like to emphasize the points I have stated in the following two items and recommend that the necessary revisions be made.

1. It seems that the word "double" should be used throughout the text instead of "dabble", as used in the summary.

2. Although it is noted that the necessary revisions have been made and highlighted in the text regarding the comments below, when the initial version and the revised version of the manuscript are compared, the sections highlighted in green and stated to be revised are exactly the same as the sections in the initial version. I guess there was an error in revising the relevant sections. I recommend that it be reviewed and the information obtained from the suggested current references mentioned in the reviewer comments be added to the discussion section.

“- The relationship between isokinetic H/Q ratio and jumping performance parameters was investigated without using RSDmod. Some recent studies are listed below. The discussion can be rearranged by reviewing these studies.

o Atik, B., Ayberk, B., Özgül, B., & Polat, M. G. (2023). The Association Between Isokinetic Strength and Strength Asymmetry and Jump Performance in Female Volleyball Players. Sport Sciences for Health, 1-8.

o Schons P, Da Rosa RG, Fischer G et al (2019) The relationship between strength asymmetries and jumping performance in professional volleyball players. Sports Biomech. 18(5):515–526.

- One of the evaluations applied in the study is a static measurement while the other is a dynamic task. Therefore, instead of a static measurement, comments can be added about the reflection of dynamic muscle strength (isokinetic) evaluations on RSDmod can be examined. Although this situation is partially mentioned at the end of the discussion, comparative comments can be made in the first parts of the discussion by giving place to case studies from the literature.”

Best Regards

Reviewer #2: All my comments were completed properly. I am satisfied with the corrections.

I would like to thank the authors for their work.

7. PLOS authors have the option to publish the peer review history of their article (what does this mean?). If published, this will include your full peer review and any attached files.

Reviewer #1: No

Reviewer #2: No

---

## [Author Response · Author response to Decision Letter 1]

20 Oct 2023

Responses to the comments

We very much appreciated your encouraging and insightful comments. We have endeavored to respond to all suggestions and comments, which further improved the understanding and potential impact of our paper. We responded to the mentioned comments in both the “revised manuscript” file and this one. In the manuscript, responses to the first reviewer have been highlighted green and yellow for the second reviewer. Hope our effort meets the editorial board's expectations. 

Sincerely Yours,

Authors.

Reviewer: 1

Firstly thank you for your effort on the revisions of the manuscript. With the revisions made, it is now in a better condition.I would only like to emphasize the points I have stated in the following two items and recommend that the necessary revisions be made.

Authors: Dear Professor, thank you for your time and consideration. 

1. It seems that the word "double" should be used throughout the text instead of "dabble", as used in the summary.

Authors: Done.

2. Although it is noted that the necessary revisions have been made and highlighted in the text regarding the comments below, when the initial version and the revised version of the manuscript are compared, the sections highlighted in green and stated to be revised are exactly the same as the sections in the initial version. I guess there was an error in revising the relevant sections. I recommend that it be reviewed and the information obtained from the suggested current references mentioned in the reviewer comments be added to the discussion section.

Authors: Dear Professor, thank you for your consideration. By using the mentioned references, the discussion section is revised. But, we added this new comment in the last paragraph of the discussion; Adding to the initial part of the discussion was a bit difficult.

Reviewer: 2

All my comments were completed properly. I am satisfied with the corrections.

I would like to thank the authors for their work.

Authors: Dear Professor, thank you for your time and consideration.

---

## [Decision Letter · Decision Letter 2]

30 Oct 2023

Is there a Relationship Between Isometric Hamstrings-to-Quadriceps Torque Ratio and Athletes' Plyometric Performance?

PONE-D-23-21774R2

Dear Dr. Babakhani,

We’re pleased to inform you that your manuscript has been judged scientifically suitable for publication and will be formally accepted for publication once it meets all outstanding technical requirements.

Kind regards,

Esedullah Akaras

Academic Editor

PLOS ONE

Additional Editor Comments (optional):

Reviewers' comments:

Reviewer's Responses to Questions

**Comments to the Author**

1. If the authors have adequately addressed your comments raised in a previous round of review and you feel that this manuscript is now acceptable for publication, you may indicate that here to bypass the “Comments to the Author” section, enter your conflict of interest statement in the “Confidential to Editor” section, and submit your "Accept" recommendation.

Reviewer #1: All comments have been addressed

Reviewer #2: All comments have been addressed

2. Is the manuscript technically sound, and do the data support the conclusions?

Reviewer #1: Yes

Reviewer #2: Yes

3. Has the statistical analysis been performed appropriately and rigorously? 

Reviewer #1: Yes

Reviewer #2: Yes

4. Have the authors made all data underlying the findings in their manuscript fully available?

Reviewer #1: Yes

Reviewer #2: Yes

5. Is the manuscript presented in an intelligible fashion and written in standard English?

Reviewer #1: Yes

Reviewer #2: Yes

6. Review Comments to the Author

Reviewer #1: Dear Authors,

Thank you for your effort on the revisions of the manuscript. With the revisions, it has become a more understandable and clear manuscript.

Best Regards

Reviewer #2: As stated previously, I have no further comments regarding this manuscript.

In my opinion, manuscrypt can be published in its current form.

7. PLOS authors have the option to publish the peer review history of their article (what does this mean?). If published, this will include your full peer review and any attached files.

Reviewer #1: No

Reviewer #2: No

---

## [Editor Report · Acceptance letter]

9 Nov 2023

PONE-D-23-21774R2 

Is there a Relationship Between Isometric Hamstrings-to-Quadriceps Torque Ratio and Athletes' Plyometric Performance? 

Dear Dr. Babakhani:

I'm pleased to inform you that your manuscript has been deemed suitable for publication in PLOS ONE. Congratulations! Your manuscript is now with our production department. 

Kind regards, 

on behalf of

Dr. Esedullah Akaras 

Academic Editor

PLOS ONE